# Usefulness of Compiled Geophysical Prospecting Surveys in Groundwater Research in the Metropolitan District of Quito in Northern Ecuador

Lilia Peñafiel [1,2], Francisco Javier Alcalá [3,4,*] and Javier Senent-Aparicio [1]

1   Departamento de Ingeniería Civil, Universidad Católica de Murcia, 30107 Murcia, Spain;
    lapenafiel@alu.ucam.edu (L.P.); jsenent@ucam.edu (J.S.-A.)
2   Empresa Pública Metropolitana de Agua Potable y Saneamiento de Quito (EPMAPS),
    Quito 17-03-0330, Ecuador
3   Departamento de Desertificación y Geo-Ecología, Estación Experimental de Zonas Áridas (EEZA-CSIC),
    04120 Almería, Spain
4   Instituto de Ciencias Químicas Aplicadas, Facultad de Ingeniería, Universidad Autónoma de Chile,
    Santiago 7500138, Chile
*   Correspondence: fjalcala@eeza.csic.es; Tel.: +34-950-281-045

**Abstract:** As in other large Andean cities, the population in the Metropolitan District of Quito (MDQ) in northern Ecuador is growing, and groundwater is becoming essential to meet the increasing urban water demand. Quito's Public Water Supply Company (EPMAPS) is promoting groundwater research for sustainable water supply, and geophysical prospecting surveys are used to define aquifer geometry and certain transient groundwater features. This paper examines the usefulness of existing geophysical prospecting surveys in groundwater research in the MDQ. A database was built using 23 representative geophysical prospecting surveys compiled from EPMAPS' public repository, official geotechnical research reports, and the scientific literature. Fifteen EPMAPS-promoted surveys used near-surface electrical techniques (seven used electrical resistivity tomography and eight used vertical electrical sounding) to explore Holocene and Pleistocene sedimentary and volcano-sedimentary formations in the 25–500-m prospecting depth range, some of which form shallow aquifers used for water supply. Four other surveys used near-surface seismic techniques (refraction microtremor) for geotechnical research in civil works. These surveys have been reinterpreted to define shallow aquifer geometry. Finally, four surveys compiled from the scientific literature used electromagnetic techniques (magnetotelluric sounding and other very low-frequency methods) to explore Holocene to late Pliocene formations, some of which form thick regional aquifers catalogued as the larger freshwater reservoirs in the MDQ. However, no geophysical prospecting surveys exploring the complete saturated thickness of the Pliocene aquifers could be compiled. Geophysical prospecting surveys with greater penetration depth are proposed to bridge this research gap, which prevents the accurate assessment of the renewable groundwater fraction of the regional aquifers in the MDQ that can be exploited sustainably.

**Keywords:** geophysical prospecting techniques; groundwater research; urban water supply; Metropolitan District of Quito; Ecuador

## 1. Introduction

The Andean Highlands roughly extend between latitudes 11° N and 8° S, are over 3000 m a.s.l., play an important role in regional freshwater supply, and are highly sensitive to climate change [1–3]. As in other high mountain areas, most ecosystem typologies are groundwater dependent [2,4–6]. The combined influence of global driving forces and some anthropogenic activities (e.g., deforestation, overgrazing, soil degradation, and water overdevelopment) is altering river flow and aquifer recharge regimes [4–8], with

negative consequences for human water supply and the preservation of dependent ecosystems [2,5,8]. Rivers and streams have traditionally been the main freshwater source to meet the water demand of downstream urban areas [4,9,10]. Increased demand in many densely populated Andean cities has driven water source diversification [2,4,6,10]. This is the case in the Metropolitan District of Quito city (MDQ) in northern Ecuador, where groundwater from the Andean Highlands and surface water transferred from the Amazonian watershed supplement traditional surface water sources [8,10]. A question arises of how is increasing groundwater extraction affecting reserves and dependent ecosystems? Applied groundwater research aimed at defining the aquifer functioning is yet incipient to answer this question accurately [11,12].

The Ecuador Water Authority and Quito's Public Water Supply Company (EPMAPS) are not immune to this problem. EPMAPS is responsible for prospecting, developing, distributing, and managing potable water in the MDQ, and promotes groundwater research to improve general hydrogeological knowledge as a prerequisite for a sustainable water supply. Hydrogeological studies use geophysical prospecting surveys to define aquifer geometry and certain transient groundwater features. Such studies typically cover two observation scales associated with two aquifer typologies. EPMAPS has used near-surface electrical geophysical techniques, such as electrical resistivity tomography (ERT) and vertical electrical sounding (VES), to explore shallow aquifer geometry and transient groundwater features required for drilling pumping wells intended to supply scattered population nuclei. The water company has also supported other public agencies and academic institutions can apply electromagnetic geophysical techniques for near-surface (very low-frequency electromagnetic methods, VLF-EMs) and deep (low-frequency magnetotelluric sounding, MTS) explorations to deduce the structure of shallow and thick regional geological formations catalogued as the larger freshwater reservoirs in the MDQ. Near-surface seismic prospecting techniques, such as refraction microtremor (REMI), have also been used to explore shallow geotechnical features.

Such geophysical prospecting techniques have proven useful in groundwater research in different hydrogeological environments [13–18]. They are non-invasive, usually inexpensive to apply, and useful when geotechnical sounding data is sparse or unable to provide subsurface information required for detailed groundwater research over multiple observation scales [15–17]. However, most geophysical prospecting surveys of interest in groundwater research are unpublished. Therefore, these experiences must be compiled and may need to be reinterpreted for groundwater research purposes. Such information concerning the aquifer saturated thickness, piezometric level, and spatial distribution of pore-water conductivity is essential to assess the fraction of groundwater that can sustainably supplement the urban water demand in the MDQ.

This paper examines the feasibility of compiled geophysical prospecting surveys in groundwater research in the MDQ, providing findings of interest to improve the hydrogeological conceptualization and identifying research gaps that can be bridged in the near future. Twenty-three representative geophysical prospecting surveys were compiled from EPMAPS' public repository, official geotechnical research reports, and the scientific literature. The compiled information was arranged in a database for interpretation. This paper does not intend to introduce new formulations, produce new data, discuss well-known principles of applied geophysical prospecting techniques, or assess the quality of the interpretations derived from the compiled surveys. This paper is organized as follows. Section 2 briefly describes the study area. Section 3 explains the steps followed to build the database. Section 4 reports the overall findings of the database analysis and gives an example of each geophysical prospecting technique. Section 5 discusses the geophysical prospecting scope, including research findings and gaps. Section 6 presents the main conclusions.

## 2. Study Area

### 2.1. Location and Climate

The MDQ is located in northern Ecuador at $0°14'$ N to $0°35'$ S and $78°10'$ W to $78°56'$ W (Figure 1), covers a surface area of 4320 km$^2$, and includes three main geomorphological sectors (Figure 1a). The 40-km-wide northern elongated Inter-Andean Valley (IAV) has an elevation ranging from 2100 to 3500 m a.s.l. and lies between the Western Andean Cordillera (WAC) (peak elevation 4776 m a.s.l., at Guagua-Pichincha Volcano) and Eastern Andean Cordillera (EAC) (peak elevation 4873 m a.s.l., at Sincholagua Volcano) (Figure 1b). The Guayllamba River flows north through the IAV and is the main surface watercourse (Figure 1b).

The MDQ exhibits a neo-tropical high-mountain climate, determined by the El Niño-Southern Oscillation and the Humboldt Current, and a steep orography [8,19,20]. Consequently, it has a marked distribution of biozones and ecosystems at different elevations including tropical mountain rainforests in the lowlands, wet alpine meadows (locally named *páramo*) in mid-slope areas, dry and cold scrublands in the highlands, and permanent snow covers at volcanoes' peaks [4,6,21].

Precipitation (P) follows a decreasing gradient from east to west, controlled by incoming Atlantic cloud fronts and elevation [22,23] and exhibits a positive gradient from low-lying areas to around 3500 m a.s.l. and a negative gradient above that elevation [19,20]. Most P occurs in February–May. In contrast, the lowest amount is recorded in July–September [23]. Annual mean P is around 1100 mm, with a coefficient of variation of 0.21 measured over the period 2003–2019. Annual mean temperature (T) is around 7.5 °C, with daily minimums in June–September and maximums in February–April. The decreasing T gradient with elevation is around 0.6–0.7 °C per 100 m elevation [21,22]. Insolation increases from low-lying areas to summits due to cloudiness induced by the Foehn effect in valleys from incoming Atlantic cloud fronts [7]. Annual mean potential evapotranspiration is around 1000 mm.

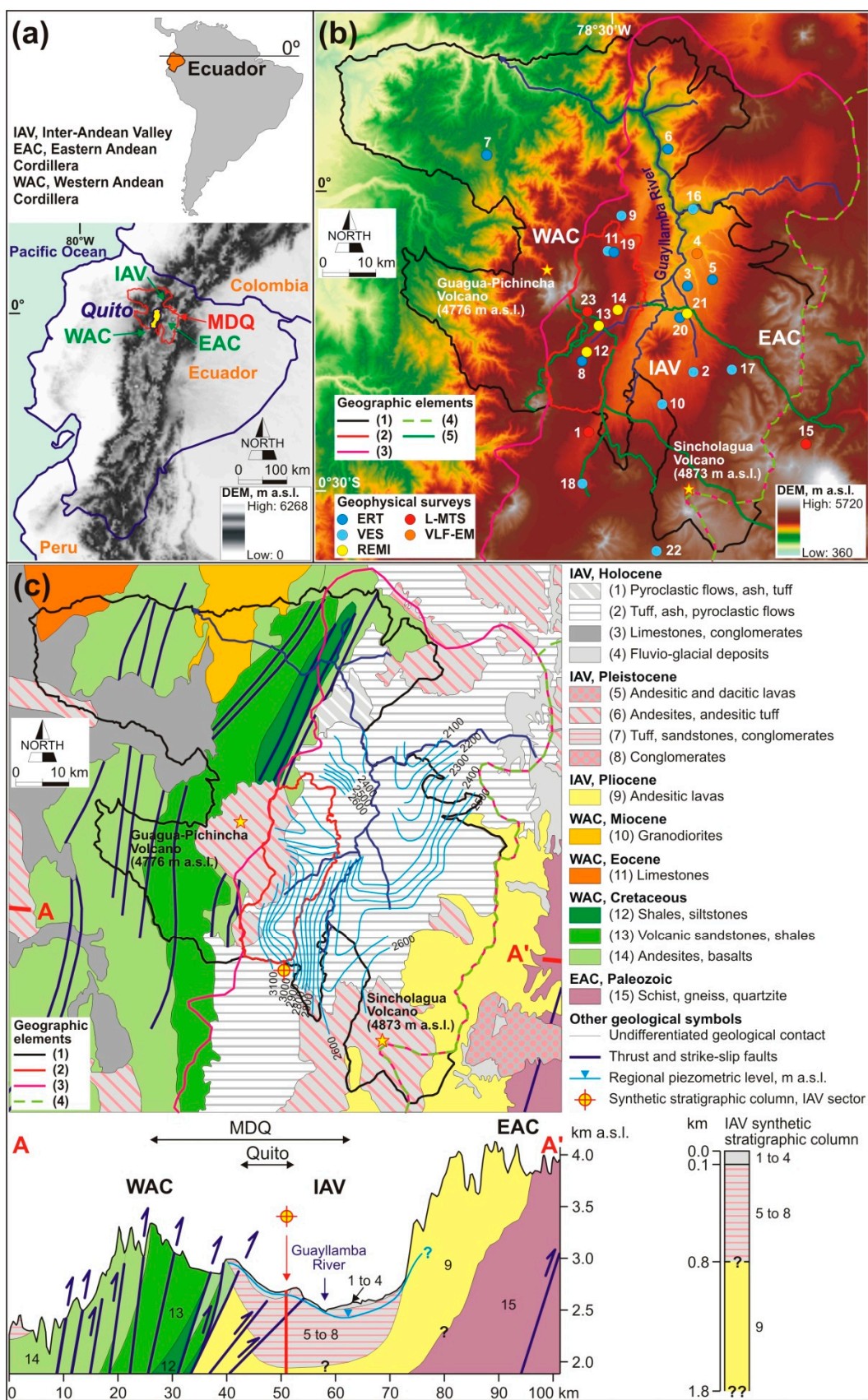

**Figure 1.** (**a**) Location of the study area. (**b**) The MDQ displayed using the 30 m-resolution Digital Elevation Model from Shuttle Radar Topography Mission (http://srtm.csi.cgiar.org/srtmdata/ (accessed on 11 February 2021), showing the

location and typology of the compiled 23 geophysical prospecting surveys and additional geographical features cited in the text: (1) MDQ, (2) Quito city, (3) Quito-Machachi Hydrogeological Unit [24–26], (4) Cayambe-Rumiñahui Hydrogeological Unit [24–26], and (5) the water transfer system used by EPMAPS to supply the MDQ [8]. ERT—Electrical Resistivity Tomography; VES—Vertical Electrical Sounding; REMI—Refraction Microtremor; L-MTS—Low-frequency Magnetotelluric Sounding; VLF-EM—Very low-frequency Electromagnetic Method. (**c**) Hydrogeological map (scale 1:250,000) of the MDQ, updated and improved from [24–26], showing regional piezometry [24–26], and the location of the hydrogeological cross-section A–A′ and a synthetic stratigraphic column of the IAV sector [18,24–26].

### 2.2. Geology and Hydrogeology

The study area belongs to the Pacific Ring of Fire, a highly active belt of volcanic and seismic activity originating from the subduction of the oceanic Nazca Plate beneath the South American Plate, which is the source of the compressive tectonics and arc magmatism of the Andes Cordillera [27,28]. The MDQ is located in the westernmost part of the NNE-trending fault-bounded Andean compressive structure, which includes the IAV between the WAC and EAC (Figure 1b).

The basement of the MDQ includes a variety of geological formations (Figure 1c). Upper Cretaceous oceanic, arc-island sequences, and volcano-sediments (codes 12–14 in Figure 1c) form the non-metamorphic basement of the WAC, which underlies Paleocene to Eocene marine turbidites and limestones (code 11 in Figure 1c), and is locally intruded by Miocene granodiorites (code 10 in Figure 1c) [29]. Subparallel belts of Paleozoic metapelitic rocks (code 15 in Figure 1c) and other volcanic-arc rocks accreted against the stable craton during the early Cretaceous form the western metamorphic basement of the EAC [30]. At present, the IAV basement depth and typology remain unknown, although Bouguer gravity anomaly data [31] would suggest an east-verging tectonic wedge of the Cretaceous WAC basement [29], which is covered by Pliocene andesitic lavas (code 9 in Figure 1c), and Pleistocene (codes 5–8 in Figure 1c) and Holocene (codes 1–4 in Figure 1c) sedimentary and volcano-sedimentary formations [32].

The IAV and EAC sectors occupy a large portion of the Quito-Machachi Hydrogeological Unit and a small part of the Cayambe-Rumiñahui Hydrogeological Unit (Figure 1b). The WAC sector is officially catalogued as a regional impervious area. In hydrogeological terms, the above geological formations can be classified into five groups attending to the permeability type and effective porosity reported in the consulted literature [25,26,33,34]: (1) the Paleozoic metapelitic EAC basement is a low-permeability formation representing the impervious lower boundary of the eastern aquifers; (2) the Late Cretaceous sedimentary and volcanic WAC basement is assumed to include low- to moderate-permeability formations comprising the impervious lower boundary of the aquifers in the western IAV sector; (3) the Pliocene and Pleistocene andesitic lavas and pyroclastic flows form thick regional compartmentalized aquifers of moderate permeability, with yield dependent upon the degree of fissuring and fracturing; (4) the Pleistocene and Holocene ash, tuff, and lahar are low- to moderate-permeability formations, often confining the above Pliocene and Pleistocene aquifers; and (5) the Pleistocene and Holocene fluvio-glacial formations form high-permeability aquifers with intergranular porosity (Figure 1c). Table 1 summarizes the compiled information regarding the permeability and effective porosity of representative geological formations in the MDQ.

Hydrogeological functioning in the MDQ depends on: (1) the low permeability of the EAC (metapelitic rocks) and WAC (sedimentary and volcanic rocks) basements; (2) the compartmentalization, thickness, and degree of fissuring and fracturing in Pliocene and Pleistocene andesitic lavas, which determine the storage capacity and permeability of these aquifers in the IAV; (3) the extent and thickness of low-permeability Pleistocene and Holocene volcano-sedimentary formations forming aquitards in the IAV; (4) the hydraulic connectivity between Pliocene and Pleistocene aquifers and between Pleistocene and Holocene aquitards, favoring the deep percolation of aquifer recharge and localized aquifer discharge; and (5) the extent and thickness of Pleistocene and Holocene coarse-grained sediments for draining runoff and aquifer discharge [9,11,12,18,26,35,36].

**Table 1.** Compiled information regarding the permeability and effective porosity of representative geological formations in the MDQ.

| Lithology | Age | MDQ Sector [1] | Permeability [2] | | Effective Porosity [3] | Reference |
|---|---|---|---|---|---|---|
| | | | Magnitude | Type | | |
| Metapelites | Paleozoic | EAC | $10^{-4}$–$10^{-2}$ (nd) | fr,fi | nd | [25,26] |
| Andesites and basalts | Cretaceous | WAC | $10^{-4}$–$10^{-2}$ (nd) | fr,fi | nd | [25] |
| Sandstones and siltstones | Cretaceous | WAC | $10^{-2}$–$10^{-1}$ (nd) | fr,fi | nd | [25] |
| Andesitic lavas | early Pleistocene | IAV | $10^{-2}$–$10^{-1}$ (0.04) | fr,fi | nd | [26] |
| Andesitic lavas | middle Pleistocene | IAV | $10^{-2}$–$10^{-1}$ (0.04) | fr,fi | 0.02–0.08 | [26,33,34] |
| Pyroclastic flows | middle Pleistocene | IAV | 0.13–0.86 (nd) | fr,fi | nd | [25,26] |
| Ash | middle Pleistocene | IAV | $10^{-3}$–$10^{-1}$ (0.01) | fr,fi | nd | [25,26] |
| Ash | late Pleistocene | IAV | $10^{-3}$–$10^{-1}$ (0.01) | fr,fi | <0.01 | [26,33,34] |
| Fluvio-glacial deposits | late Pleistocene | IAV | 0.05–10 (1.02) | fi,ip | 0.01–0.03 | [25,26] |
| Ash | Holocene | IAV | $10^{-3}$–$10^{-1}$ (nd) | fr,fi | nd | [25,26] |
| Avalanche flows | Holocene | IAV | $10^{-2}$–$10^{-1}$ (nd) | fr,fi | nd | [25,26] |
| Lahar | Holocene | IAV | $10^{-3}$–$10^{-2}$ (0.01) | fr,fi | 0.01–0.06 | [26,33,34] |
| Alluvial | Holocene | IAV | 0.05–0.18 (0.12) | ip | 0.05–0.12 | [25,26] |
| Glacier and moraines | Holocene | IAV | 0.05–0.15 (0.09) | ip | 0.05–0.15 | [25,26] |

[1] EAC—Eastern Andean Cordillera, WAC—Western Andean Cordillera, and IAV—Inter-Andean Valley. [2] Permeability in m d$^{-1}$; magnitude refers to theoretical ranges and experimental values after borehole surveying (in parenthesis); fr—fracturation, fi—fissuration, and ip—intergranular porosity. [3] Effective porosity as a fraction; magnitude refers to experimental values after borehole surveying. nd—no data.

### 2.3. Urban Water Demand

Quito has historically been supplied from local rivers and streams. Since the 1990s, internal migration has produced rapid population growth in the MDQ, leading to increased water demand and the need to diversify water sources [37]. The MDQ currently has around 2.7 million inhabitants. Groundwater from the highlands (the EAC sector) and surface water transferred from the Amazonian watershed supplement the traditional surface water sources [8,10]. Groundwater exploitation began in the 1960s when the first pumping wells were drilled to supply northern urban districts [37]. Since then, EPMAPS has drilled more than 120 pumping wells to supply the increasing water demand [26]. The water supply system currently covers about 99% of the inhabitants, making the MDQ one of the best-served urban areas in Latin America. Groundwater meets around 16% of the total urban water demand. This figure will undoubtedly increase due to the noticeable population increase projected for the period 2020–2040 [37].

### 3. Data Compilation

A data search was conducted to examine the feasibility of existing geophysical prospecting surveys in groundwater research in the MDQ. The rationale was to create a database to cover as many geological formations (preferably those catalogued as aquifers), research interests, and prospecting techniques as possible. The selection prioritized geophysical surveys that explored depths of at least 10 m and used external validation data, such as geotechnical soundings logs and/or additional prospecting techniques. The selection also considered those surveys developed or promoted by the EPMAPS in sites where it has (or intends to have) operative water catchments. Therefore, EPMAPS' public repository (information available on request), official geotechnical research reports, and the scientific literature were consulted. Finally, 23 representative surveys covering the abovementioned scopes and priorities were selected to build the database in Table 2. Most of the compiled surveys were performed in the IAV sector (Figure 1b).

**Table 2.** Database comprising the compiled 23 geophysical prospecting surveys of interest in groundwater research in the MDQ. The data are clustered according to the applied geophysical prospecting technique, explored aquifer typology, deduced transient groundwater features, and additional technical information.

| ID | Coordinates | Elevation, m a.s.l. | Geophysical Technique [1] | | | Geological Environment [2] | | | | | | Research Interest [3] | | | Additional Technical Information [4] | | | | | | Reference |
|---|---|---|---|---|---|---|---|---|---|---|---|---|---|---|---|---|---|---|---|---|---|
| | | | T1 | T2 | T3 | G1 | G2 | G3 | G4 | G5 | G6 | R1 | R2 | R3 | Variable | AQF | AQT | Profiles | PL | PD | |
| 1 | 78°32′ W 0°24′ S | 3046 | | | b | a,b | a,b,c | a,b,c | c | a | | a,b | b | a | ER, Ω m | 10–210 | 220–8010 | 1 | 1300 | 1800 | [18] |
| 2 | 78°22′ W 0°18′ S | 2644 | a | | | a,b | a,b | | a,b,c | a | | a | b | b | ER, Ω m | 62–170 | | 10 | 400–500 | 250 | [38] |
| 3 | 78°22′ W 0°09′ S | 2379 | b | | | b | | a,b,c | a,b,c | a | | a | b | b | ER, Ω m | 80–150 | | 7 | 504–855 | 70–160 | [39] |
| 4 | 78°21′ W 0°06′ S | 2350 | | | a | b | a,b,c | | a,b,c | a | | a | b | b | ER, Ω m | 15–195 | 210–280 | 13 | 160–750 | 130 | [39] |
| 5 | 78°20′ W 0°09′ S | 2484 | b | | | b | a,b,c | | a,b,c | a | | a | b | a,b | ER, Ω m | 30–75 | 210–250 | 2 | 880 | 137 | [40] |
| 6 | 78°24′ W 0°04′ N | 2064 | b | | | b | a,b,c | | b,c | a | | a,b | a | a,b | ER, Ω m | 10–50 | | 2 | 303–358 | 50 | [40] |
| 7 | 78°42′ W 0°03′ N | 1823 | b | | | a,b | a,b | | b,c | | a | a | a | a,b | ER, Ω m | 17–198 | 210–315 | 1 | 715 | 120 | [40] |
| 8 | 78°33′ W 0°17′ S | 2860 | b | | | a,b | a,b,c | a,b | | | | a | a | c | ER, Ω m | 3–210 | 215–300 | 9 | 250 | 40 | [41] |
| 9 | 78°29′ W 0°00′ N | 2736 | a | | | a,b | a,b | b,c | c | | | a | b | a | ER, Ω m | 20–40 | | 9 | 600–1000 | 322 | [42] |
| 10 | 78°25′ W 0°21′ S | 2690 | a | | | a | | a,b,c | a,b,c | | | a | a | a,b | ER, Ω m | 17–28 | 215–345 | 9 | 600–1000 | 271 | [42] |
| 11 | 78°30′ W 0°06′ S | 2722 | a | | | a,b | a,b,c | a,b,c | a,b,c | | | b | b | a,b | ER, Ω m | 30–98 | | 11 | 400–1000 | 200–500 | [43] |
| 12 | 78°32′ W 0°16′ S | 2849 | | a | | a,b | a,b,c | a,b,c | | | | a | a | c | VS, m s$^{-1}$ | 95–680 | 95–680 | 171 | 8600 | 40–55 | [44] |
| 13 | 78°31′ W 0°13′ S | 2826 | | a | | a,b | a,b,c | c | | | | b | b | c | VS, m s$^{-1}$ | 135–1050 | 135–1050 | 15 | 3200 | 120 | [44] |
| 14 | 78°29′ W 0°12′ S | 2777 | | a | | a,b | a,b,c | a,b,c | | | | a | b | c | VS, m s$^{-1}$ | 125–710 | 125–710 | 171 | 10,330 | 40–55 | [44] |
| 15 | 78°11′ W 0°25′ S | 4184 | | | b | a,b | | c | c | a | b | b | b | d | ER, Ω m | 30–215 | 230–3190 | 130 | 15,000 | 4000 | [45] |
| 16 | 78°22′ W 0°02′ S | 2145 | a | | | a | | | a,b,c | a | | a | b | a | ER, Ω m | 50–170 | 218–457 | 23 | 600–1000 | 230 | [46] |
| 17 | 78°18′ W 0°18′ S | 3260 | a | | | a | a,b | c | c | a | | b | b | a | ER, Ω m | 28–56 | | 3 | 600–800 | 180 | [47] |
| 18 | 78°33′ W 0°29′ S | 2835 | a | | | a,b | a,b,c | a | a,b,c | a | | b | b | b | ER, Ω m | 45–150 | 255–400 | 16 | 1000 | 200 | [48] |
| 19 | 78°29′ W 0°06′ S | 2693 | b | | | a,b | a,b | a,b | | | | a | a | a,b | ER, Ω m | 40–100 | 220–320 | 3 | 715 | 60 | [49] |
| 20 | 78°22′ W 0°12′ S | 2280 | b | | | a | | | a,b,c | | | a | a | c | ER, Ω m | 20–150 | | 3 | 110 | 25 | [50] |
| 21 | 78°22′ W 0°12′ S | 2400 | | a | | a | | a,b,c | | | | a | a | c | VS, m s$^{-1}$ | | 720–945 | 7 | 120 | 60 | [50] |
| 22 | 78°25′ W 0°35′ S | 3743 | a | | | | | a,b,c | c | a | | b | a | a | ER, Ω m | 56–203 | 225–850 | 20 | 1000 | 150 | [51] |
| 23 | 78°30′ W 0°12′ S | 2800 | | | b | a | a,b,c | b,c | c | a | a | b | b | a | ER, Ω m | 22–230 | 235–27,100 | 3 | 17,000 | 1500 | [52] |

[1] T1—Near-surface electrical techniques: (a) VES and (b) ERT. T2—Near-surface seismic techniques: (a) REMI. T3—Electromagnetic techniques: (a) VLF-EM and (b) Low-frequency MTS. [2] G1—Holocene sedimentary and volcano-sedimentary formations: (a) anthropogenic fillings and soils, and (b) silty and sandy ash. G2—late Pleistocene sedimentary and volcano-sedimentary formations: (a) ash and pumice ash, (b) tuff and paleo-soils, and (c) mud flows. G3—early–middle Pleistocene sedimentary and volcano-sedimentary formations: (a) lacustrine deposits, paleo-soils, peats, tuff, and microbreccias; (b) ash, pyroclastic flows, and tuff; and (c) avalanche flows, volcanic breccias and lavas. G4—early Pleistocene sedimentary and volcano-sedimentary formations: (a) alluvial sand and conglomerates; (b) siltstones and tuff; and (c) lahar, andesitic lavas, and avalanche flows. G5—late Pliocene volcanic formations: (a) andesitic lavas. G6—Basement: (a) Cretaceous WAC and IAV basement, and (b) Paleozoic EAC basement. [3] R1—Aquifer geometry: (a) layer thickness, and (b) fissuring and fracturing. R2—Aquifer dynamics and functioning in natural regime: (a) first groundwater observation, (b) regional piezometric level, (c) high-conductivity areas, and (d) geothermal areas. R3—Other applications: (a) basic research, (b) water supply, (c) civil works, and (d) geothermal energy. [4] AQF—Range of the variables electrical resistivity (ER) and share-wave velocity (VS) in geological formations catalogued as aquifers. AQT—Range of the variables ER and VS in geological formations catalogued as aquitards. PL—Prospecting length of the profiles in m. PD—Prospecting depth of the profiles in m.

Survey information was catalogued according to: (1) the applied prospecting technique; (2) the explored aquifer typology; (3) deduced transient groundwater features; and (4) additional technical information, such as the magnitude of the geophysical variables in geological formations catalogued as aquifers and aquitards, and prospecting length and depth (Table 2). The compiled data were initially checked to ensure a suitable statistical sample of the range of field technical conditions allowed by each geophysical technique. In all surveys, the original interpretations were examined to (i) standardize the age, lithological description, and hydrogeological behavior of the geological formations; (ii) adapt the achieved transient groundwater features to the scope of this paper; and (iii) adjust the drawing style to the scientific editing requirements. In the REMI and low-frequency MTS surveys, the original geotechnical and geodynamic findings were reinterpreted for groundwater research purposes.

Section 4 examines and classifies the compiled 23 surveys into three main groups of techniques (electrical, seismic, and electromagnetic) used to explore two main aquifer typologies (shallow and regional) and deduce two main research interests (aquifer geometry and transient groundwater features). This section also provides a representative survey of each technique. This survey represents an average condition of (i) prospecting depth and length, (ii) prospected geological formations, and (iii) deduced hydrogeological features regarding the compiled surveys of each technique.

## 4. Results

### 4.1. Near-Surface Electrical Surveys

Near-surface electrical techniques take voltage measurements between two potential electrodes installed on the land surface once direct current is injected into two current electrodes. Such techniques allow the calculation of subsurface electrical resistivity (ER) [Ω m], reciprocal of subsurface electrical conductivity (EC). Penetration depth and resolution depend on subsurface EC, which is a function of transient pore-water EC and steady ground EC, the input voltage used, and the electrode spacing adopted [53–55].

Fifteen surveys used near-surface electrical techniques; of these, eight surveys used VES for 1D ER models, and seven used ERT for 2D ER models. The VES surveys were part of groundwater research technical reports [38,42,43,46–48,51], whereas the ERT surveys included groundwater research technical reports [40,41,49,50] and scientific documents [39] (Table 2). The VES surveys aimed to define the punctual thickness of Holocene and late Pleistocene shallow aquifers. The ERT surveys aimed to explore the geometry of shallow aquifers and transient groundwater features, such as first groundwater observation, regional piezometric level, and pore-water conductivity. Both the VES and ERT surveys used Schlumberger, Wenner, and dipole–dipole arrays as the typical electrode configuration. The prospecting length range was 400–1000 m for the VES and 110–880 m for the ERT surveys, and the prospecting depth range was 150–500 m for the VES and 25–160 m for the ERT surveys (Table 2). For the geological formations catalogued as aquifers, the ER range was 17–203 Ω m for VES and 3–150 Ω m for ERT. For the geological formations catalogued as aquitards, the ER range was 215–850 Ω m for VES and 210–320 Ω m for ERT.

The ERT survey labelled 19 in Figure 1b and Table 2 was selected (Figure 2). This survey was performed in January 2016 and included three NNE–SSW ERT profiles (Figure 2b) covering a total prospecting length of 715 m [49]. The survey was part of groundwater research promoted by EPMAPS to supply scattered areas in the northern district of Quito city. Research interests were the geometry of shallow geological formations (some forming aquifers) and transient groundwater features, such as first groundwater observation.

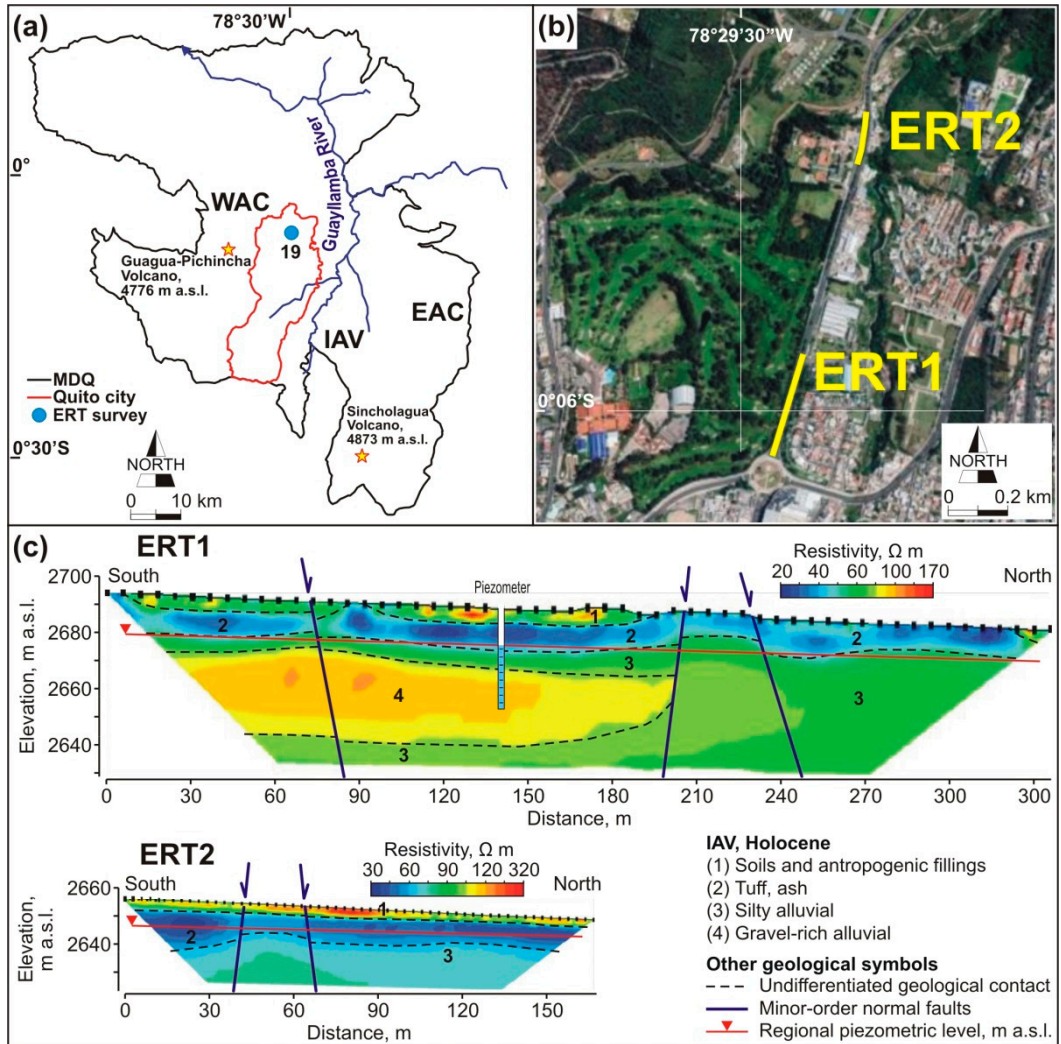

**Figure 2.** (**a**) General location of the selected ERT survey labelled 19 in Figure 1b and Table 2, updated and reinterpreted from Yautibug and Herrera [49]. (**b**) Detailed location of two selected ERT profiles, here called ERT1 and ERT2. (**c**) ERT1 and ERT2 profiles. The first groundwater observation (in this survey equivalent to the regional piezometric level) is singled out. Hydrogeological reinterpretation of ER models after [49] and local hydrogeological information [26,33,34]. Profiles are topographically corrected, and the vertical-to-horizontal scale ratio is 1:1.

The ER data were acquired using the SuperSting R8/IP eight-channels and the SuperSting R1/IP single-channel Memory Earth Resistivity and IP Meter by Advanced Geosciences Inc., Austin, TX. Fifty-six electrodes were placed along each ERT profile using a variable 3–6-m spacing and applying an input voltage of 200 V. A Schlumberger electrode array was used. See Yautibug and Herrera [49] for further methodological details.

Figure 2b shows the location of selected ERT profiles, here called ERT1 and ERT2. The profile features were prospecting lengths of 330 (ERT1) and 165 m (ERT2), prospecting depths of 59 (ERT1) and 28 m (ERT2), ER in the ranges of 20–175 (ERT1) and 25–330 Ω m (ERT2), and average root-mean-square errors (RMSE) of 1.80 (ERT1) and 2.94 (ERT2). Both ERT1 and ERT2 exposed quite similar horizontal and vertical ER distributions (Figure 2c). The ER values were typical of sedimentary and volcano-sedimentary rocks with a variable degree of saturation [56].

Local hydrogeological information [26,33,34] was used to reinterpret the ER models. From top to bottom, the vertical ER distribution was as follows: (i) 1–5 (ERT1) and 1–8 m (ERT2) of discontinuous porous soils and anthropogenic fillings with ER in the 70–300 Ω m range; (ii) 3–5 (ERT1) and 6–15 m (ERT2) of tuff and ash formation catalogued as aquitard

with ER in the 20–50 Ω m range; (iii) 5–10 (ERT1) and 10–20 m (ERT2) of silty alluvial formation catalogued as aquitard with ER in the 40–80 Ω m range; (iv) 5–50 (ERT1) and 10–50 m (ERT2) of coarse-grained alluvial formation catalogued as aquifer with ER in the 80–120 Ω m range; and (v) 5–10 m (ERT1) of silty alluvial formation catalogued as aquitard with ER in the 40–80 Ω m range. The interbedded coarse-grained alluvial formation between the above silty alluvial formation (which includes items iii and v) is part of a shallow aquifer that provided the first groundwater observation corresponding to the regional piezometric level. Changes in the thickness and spatial continuity in the vertical ER distribution are due to sedimentary processes (e.g., lateral facies changes and erosive channels) and the action of minor-order normal faults described in the area [26,33,34].

### 4.2. Near-Surface Seismic Surveys

Near-surface seismic techniques respond to the steady shear modulus of subsurface materials, expressing seismic shear-wave velocity (VS) [L T$^{-1}$], in which the Rayleigh wave fundamental mode dispersion curve and higher modes (if present) are extracted from a shot record and then inverted to generate 1D VS models [57–62]. A succession of geophones records ambient microtremor to generate the Rayleigh waves from which a 2D VS model is obtained [61,62].

REMI was the near-surface seismic technique used to acquire VS data and map 2D VS models in four surveys designed to support geotechnical research in civil works (Table 2). The total prospecting length was 22.2 km, the prospecting depth was 40–120 m, and VS varied in the 95–1050 m s$^{-1}$ range (Table 2).

This paper reinterpreted the 2D VS models for the shallow geological definition following the interpretative criteria reported by Paz et al. [63] and Alcalá et al. [64]. These authors propose that subsurface VS propagation is a site-specific steady property determined by effective compaction and therefore is dependent on the age and depth of each geological material piled vertically [65–68]. The different relationships between VS and age and depth in different lithologies described in the scientific literature [63–71] were used to reinterpret the VS models.

The VS increased in depth according to the increasing age and compaction of geological materials, from less than 200 m s$^{-1}$ in recent anthropogenic fillings and lacustrine formations, 200–550 m s$^{-1}$ in Holocene sedimentary and volcano-sedimentary formations, and more than 550 m s$^{-1}$ in Pleistocene volcano-sedimentary formations. As in other near-surface seismic techniques, REMI cannot disambiguate boundaries of different geological formations with similar VS [63,64,70,71]. This limitation to make inner divisions was solved by using external validation data, such as regional [24,25] and local [26,33,34] geological information, geotechnical soundings logs, and other prospecting techniques [63,64].

The REMI surveys labelled 12 and 14 in Figure 1b and Table 2 were selected (Figure 3). They were performed in November 2011 as part of the Quito Subway geotechnical research, which included 201 REMI profiles grouped into three REMI surveys with prospecting lengths of 8.6 (southern Quito), 3.2 (central Quito), and 10.3 km (northern Quito) [44]. Since these prospecting lengths are too long to be drawn in detail, two 2.5-km sections from the southern- and northern-Quito REMI surveys exploring the most representative geological formations were selected (Figure 3b).

The VS data were acquired using the DAQlink-4 24-channels Compact Seismograph and the 4.5 Hz Geo-Space geophones by Seismic Source Co., Ponca city, OK, USA. The following configuration was applied: a recording array of 24 vertical component geophones, 4-m geophone spacing for a prospecting depth of around 40 m, 10-m displacement between readings, and a sampling rate of 0.25 m s$^{-1}$. See Cataldi [44] for further details about the data processing and mathematical inversion.

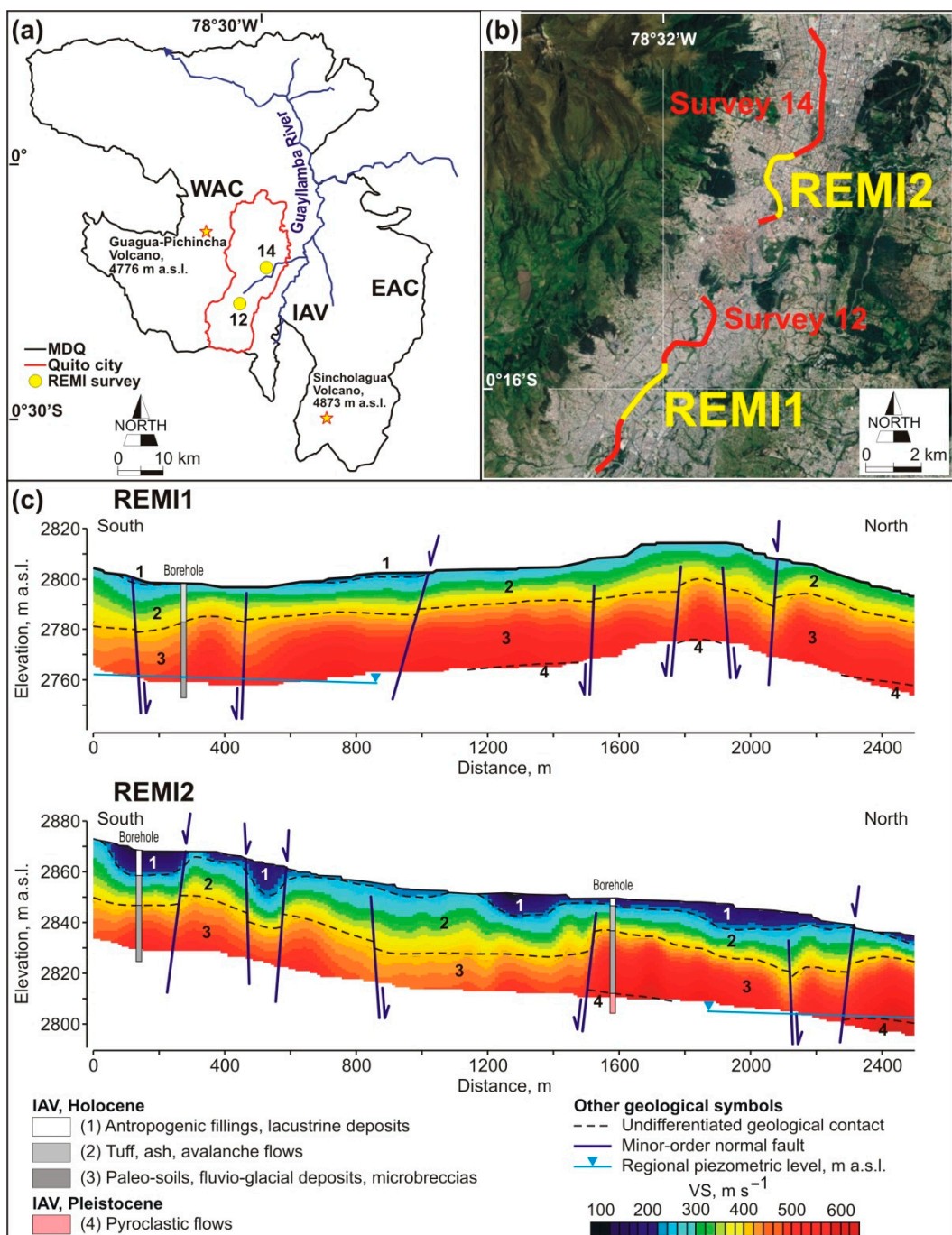

**Figure 3.** (**a**) General location of the selected southern- and northern-Quito REMI surveys labelled 12 and 14 in Figure 1b and Table 2, updated and reinterpreted from Cataldi [44]. (**b**) Detailed location of two selected 2.5-km sections from REMI surveys 12 and 14, here called REMI1 and REMI2. (**c**) REMI1 and REMI2 profiles' sections. The first groundwater observation (in these surveys equivalent to the regional piezometric level) is singled out. Hydrogeological reinterpretation of the two VS models after [44] and regional [24,25] and local [26,33,34] hydrogeological information. Profiles are topographically corrected, and the vertical-to-horizontal scale ratio is 1:0.13.

Figure 3b shows the location of the two selected REMI surveys sections, here called REMI1 from survey 12 and REMI2 from survey 14. The section features comprised an imposed prospecting length of 2.5 km, a prospecting depth of 40 m, VS in the range of 120–580 (REMI1) and 190–610 m s$^{-1}$ (REMI2), and average RMSEs of 10.83 (REMI1) and 9.42 (REMI2) (Figure 3c). The VS values were typical of sedimentary [63,64] and volcano-sedimentary [72] rocks.

Based on geotechnical soundings logs and regional [24,25] and local [26,33,34] geological information, the vertical VS distribution was reinterpreted from top to bottom as follows: (i) 5–15 (REMI1) and 1–5 m (REMI2) of anthropogenic fillings, soils, and lacustrine deposits with VS less than 200 m s$^{-1}$; (ii) 10–40 (REMI1) and 10–20 m (REMI2) of Holocene tuff, ash, and avalanche flows with VS in the 200–400 m s$^{-1}$ range; (iii) 1–40 m (REMI1) and less than 10 m (REMI2) of Holocene paleo-soils, fluvio-glacial deposits, and microbreccias with VS in the 400–550 m s$^{-1}$ range; and (iv) 10–50 (REMI1) and 10–30 m (REMI2) of Pleistocene pyroclastic flows with VS higher than 550 m s$^{-1}$. The Pleistocene formations were only occasionally identified. The horizontal continuity of the vertical VS distribution was frequently interrupted by sedimentary processes (e.g., lateral facies changes and erosive channels) and the action of minor-order normal faults described in the urban area of Quito city [26,33,34].

### 4.3. Electromagnetic Surveys

Four surveys used electromagnetic techniques, specifically VLF-EM and low-frequency MTS, for the geometry and structure of Holocene to late Pliocene formations resulting from the action of first-order thrusts and strike-slip faults [18,39,45,52]. In general, electromagnetic techniques infer subsurface ER from measurements of natural geomagnetic and geoelectric field variations at the ground surface [73,74]. In particular, VLF-EM and low-frequency MTS use a fixed grounded dipole or horizontal loop as an artificial signal source to determine ER [75,76]. Both natural and controlled-source electromagnetic signals are used to obtain 1D ER models beneath the measurement site [77]. The ER dataset at the corresponding depths and signal-source distances are mathematically inverted to produce a 2D ER model [75,77].

One survey used VLF-EM (<10 Hz) in 13 profiles with a prospecting length in the 160–750 m range, a maximum prospecting depth of 130 m, and ER in the 15–280 Ω m range [39]. Three surveys used low-frequency MTS (>10 Hz) [18,45,52] with a total prospecting length of 33.3 km, a maximum prospecting depth of 4000 m, and ER in the ranges of 10–220 Ω m for geological formations catalogued as aquifers and 220–27,100 Ω m for geological formations catalogued as aquitards (Table 2).

The low-frequency MTS survey labelled 01 in Figure 1b and Table 2 was selected (Figure 4). This survey was performed in 2016 and included 13 measurement sites aligned in an NNE–SSW profile perpendicular to the Saguanchi Gorge strike-slip fault in the southern border of the MDQ (Figure 4b) [18]. Strike-slip faulting produces additional extensional areas disposed perpendicular to the primary shortening tectonic component evidenced by the Quito Fault System, which has implications for the drainage network and the extension and thickness of aquifers in the IAV [18].

The ER data were acquired using the StrataGem EH-4 four-channels Hybrid Source with a TxIM2 transmitter and electric BE-26 and magnetic G100k sensors by Geometrics, Inc., San Jose, CA, USA. The configuration was as follows: 13 measurement sites were set up using a variable 40–170 m array spacing and applying three frequencies in the 10–50 Hz range to record ER from depths of 0.6 to 1.8 km. The recorded ER data were mathematically inverted to obtain a 2D ER model. See Peñafiel et al. [18] for further methodological details.

Figure 4b shows the location of the selected low-frequency MTS profile, here called MTS1 (Figure 4c). The profile features included a prospecting length of 1300 m, a prospecting depth of 1800 m, ER in the 10–8010 Ω m range, and an average RMSE of 14.32. The ER values were similar to that reported for similar volcano-sedimentary rocks with variable degrees of fissuring, fracturing, and saturation [52,56].

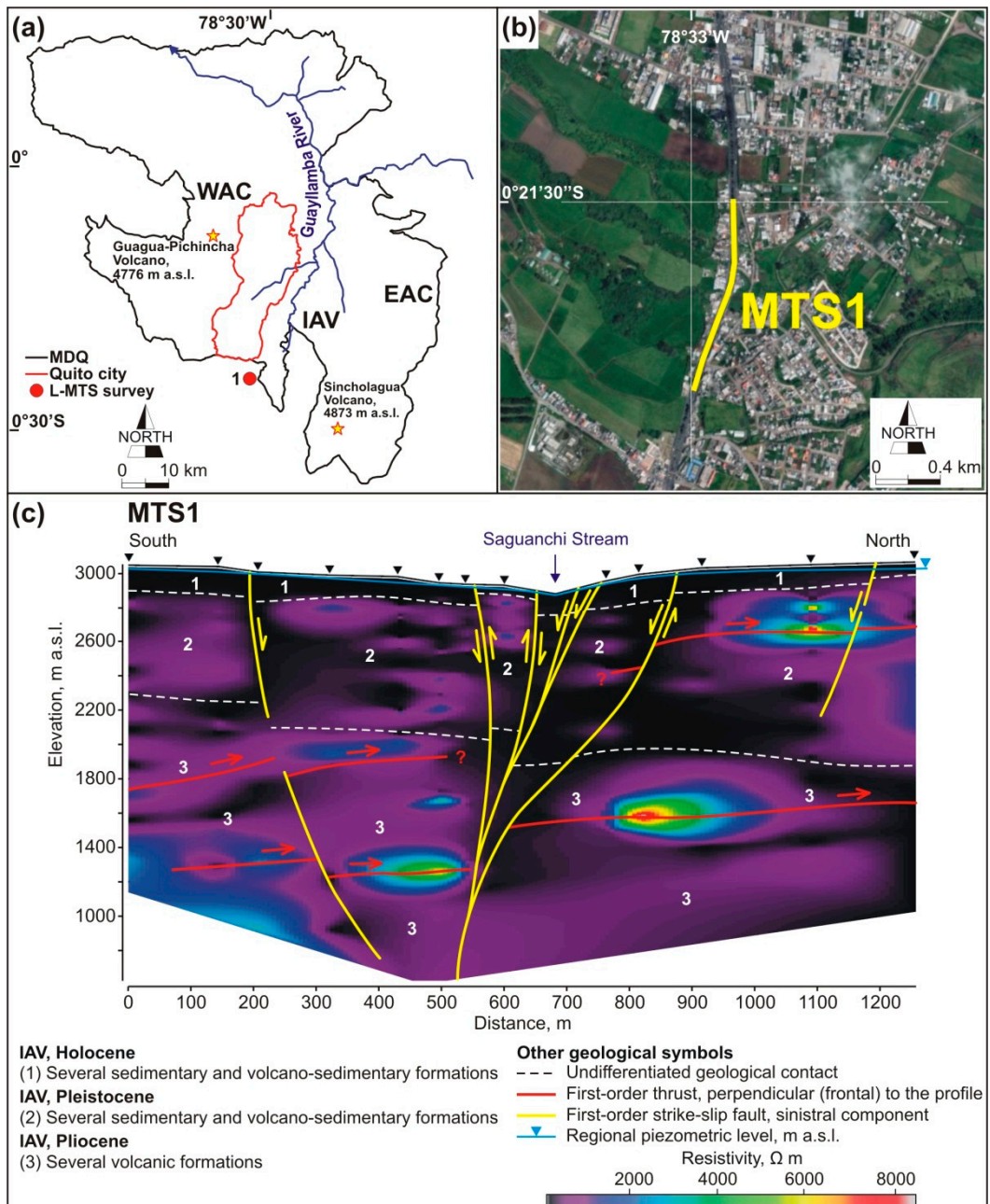

**Figure 4.** (**a**) General location of the selected low-frequency MTS survey labelled 1 in Figure 1b and Table 2, updated and reinterpreted from Peñafiel et al. [18]. (**b**) Detailed location of the selected low-frequency MTS profile, here called MTS1. (**c**) MTS1 profile. The regional piezometric level is singled out. Hydrogeological reinterpretation of the ER profile after [18] and regional [24,25] and local [26,33,34] hydrogeological information. Profile is topographically corrected, and the vertical-to-horizontal scale ratio is 0.34:1.

The ER model displayed (i) high-resistivity southern and northern sectors with an average ER around 1000 $\Omega$ m and several anomalies higher than 3000 $\Omega$ m, and (ii) a low-resistivity central sector bounded by strike-slip faults with an average ER lower than 1000 $\Omega$ m. In detail, ER values in the 10–50 $\Omega$ m range are associated with (i) strongly fractured volcanic rocks of different ages and high saturation degrees within the strike-slip fault zone, and (ii) sub-horizontal contacts within the Pleistocene and late Pliocene formations inferred by first-order thrusts observed at other sites (Figure 4c). ER values in the 50–200 $\Omega$ m range are associated with Holocene and Pleistocene volcano-sedimentary formations with moderate degrees of fissuring, fracturing, and saturation. ER values in

the 200–1000 Ω m range are attributed to Pleistocene and late Pliocene volcanic formations with moderate fissuring and fracturing and low to moderate degrees of saturation. ER values higher than 1000 Ω m are interpreted as Pleistocene and late Pliocene volcanic rocks with low fissuring, fracturing, and saturation degrees.

## 5. Discussion

For the period 2020–2040, climate change projections foresee declining surface water sources in the Andean highlands, while the population in the MDQ could increase from 2.7 to about 4.2 million inhabitants [6,8,10]. The consequence is more groundwater exploitation to supplement the increasing urban water demand [37]. Water Authority of Ecuador and EPMAPS are aware of this problem and have already begun to promote applied groundwater research for sustainable use. Geophysical prospecting surveys can contribute to improve the hydrological conceptualization. However, the existing geophysical surveys explored different observations scales aimed to cover different research interests. So, the geophysical information must be examined and integrated before use in groundwater research. Most geophysical surveys have been performed in the most populated IAV sector (Figure 1b, Table 2), where groundwater exploitation is concentrated and signs of degradation have been reported [36,78].

To examine the usefulness of the subsurface geophysical information in groundwater research in the IAV, the area covered by each geophysical prospecting survey (defined by the corresponding prospecting length and depth) (Table 2) was superimposed onto a synthetic stratigraphic column in the southern border of the MDQ (Figure 1c). Of the compiled 23 geophysical prospecting surveys (Table 2), only the 20 ones located in the IAV (Figure 5) were selected and classified into three methodological groups (electrical, seismic, and electromagnetic) covering two observation scales and two aquifer typologies: shallow Holocene and late Pleistocene aquifers and thick regional middle–early Pleistocene and late Pliocene aquifers.

The VES and ERT surveys were used to define the geometry of shallow Holocene and late Pleistocene aquifers, deduce the regional piezometric level, and qualify pore-water conductivity. The prospecting depth was 150–500 m for the VES and 25–165 m for the ERT surveys (Figure 5). Reinterpretation of the 2D ER models shows that the ER range was 17–203 Ω m for VES and 3–150 Ω m for ERT in those geological formations catalogued as aquifers. These figures agree with the expected EC variability in saturated media associated with variable contributions of natural (e.g., recharge, thermalism, mineral dissolution) and anthropogenic (e.g., domestic, agriculture, industry) salinity sources. Groundwater conductivity deduced in shallow aquifers was higher than in thick regional aquifers. In those geological formations catalogued as aquitards, the ER range was 215–850 Ω m for VES and 210–320 Ω m for ERT. These figures agree with the expected lower variability of conductivity induced by the homogeneous clay content and barely variable lower pore-water content. The regional piezometric level varied depending on the aquifer hydraulic functioning, explored aquifer zone (recharge, transit, and discharge), and topography (Table 2).

REMI surveys were originally performed in geotechnical research for civil works. The 2D VS models were reinterpreted for the geometric definition of shallow Holocene and late Pleistocene aquifers, which is an innovative research application. The prospecting depth was 40 m (Figure 5). The VS values were less than 200 m s$^{-1}$ in recent anthropogenic fillings, soils, and lacustrine formations; 200–550 m s$^{-1}$ in Holocene formations; and more than 500 m s$^{-1}$ in late Pleistocene formations.

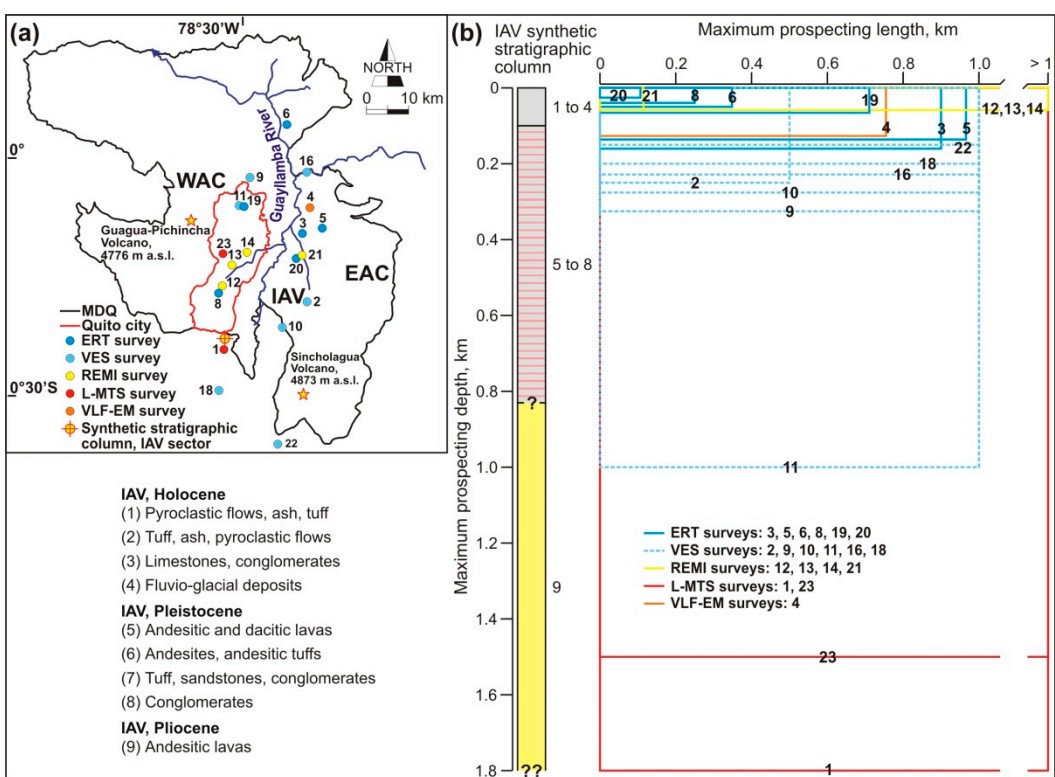

**Figure 5.** (**a**) General location of the 20 geophysical prospecting surveys in the IAV sector as shown in Figure 1b and Table 2; surveys 15 and 17 in the EAC sector and survey 7 in the WAC sector are excluded from this analysis. (**b**) A synthetic stratigraphic column of the IAV sector, as shown in Figure 1c. The typology and area covered by each geophysical prospecting survey are indicated. Acronyms ERT, VES, REMI, L-MTS, and VLF-EM are defined in Figure 1b.

VLF-EM and low-frequency MTS surveys provided the geometry and structure of Holocene to late Pliocene formations resulting from the action of first-order thrusts and strike-slip faults. The prospecting depths were 130 m for the VLF-EM survey and 1500, 1800, and 4000 m for the three low-frequency MTS surveys, of which only the first two were in the IAV sector (Figure 1b, Table 2). The hydrogeological reinterpretation of these two low-frequency MTS surveys [18,52] provided two significant findings: (i) the delineation of first-order thrusts and strike-slip faults controlling the geometry and stacking structure of Holocene to late Pliocene formations; and (ii) the identification of hitherto unknown disconnections (evidenced as high-resistivity fringes) between aquifers (evidenced as low-resistivity spaces) previously defined as hydraulically connected [26,36], resulting in less groundwater storage than previously known. An example is given in Figure 4c.

Despite the geophysical prospecting findings (Figure 5), three relevant gaps limiting a suitable hydrogeological conceptualization in the IAV sector still remain: (i) identifying the complete saturated thickness of Pliocene formations; (ii) elucidating the existence of older underlying Neogene formations of unknown hydrogeological behavior; and (iii) defining the IAV basement depth and typology, which is assumed to be equivalent to the WAC basement, after Bouguer gravity wedge data [31]. These gaps should be the subject of future research. Geophysical prospecting surveys with greater penetration depth could provide this basic aquifer information to assess the groundwater resource of Pleistocene and Pliocene andesitic lavas catalogued as the larger freshwater reservoirs in the MDQ.

## 6. Conclusions

The MDQ is a sparse-data area where definition of shallow and thick regional aquifers functioning, as well as their hydraulic relationships, is yet incipient. Different geophysical prospecting surveys originally devoted to different research interests can be integrated

to provide subsurface information of interest in groundwater research. However, the published geophysical information is restricted to some research papers and scientific documents that aimed to investigate the transient groundwater features of shallow aquifers and the structure of deep (but not the deepest) geological formations. The compilation and examination of unpublished geophysical prospecting surveys contribute to improving the hydrogeological conceptualization, as well as to proposing additional research to bridge important gaps, which prevents the implementation of robust hydrological numerical tools to assess the groundwater resource.

A data search was conducted to examine the feasibility of existing geophysical prospecting surveys in groundwater research in the MDQ. Sources of information were the EPMAPS' public repository for near-surface electrical techniques (ERT and VES surveys), official geotechnical research reports in civil works for near-surface seismic techniques (REMI surveys), and scientific documents for electromagnetic techniques (MTS surveys). Finally, 23 representative geophysical prospecting surveys were compiled. Most of the surveys were performed in the IAV sector, where groundwater exploitation is concentrated. The ERT and VES surveys explored aquifer geometry and transient groundwater features of Holocene and late Pleistocene formations (some forming shallow aquifers), such as the aquifer saturated thickness, piezometric level, and spatial distribution of pore-water conductivity. The REMI surveys were reinterpreted to deduce the geometry of Holocene formations and, occasionally, late Pleistocene formations. The VLF-EM and low-frequency MTS surveys provided the structure of Holocene to late Pliocene formations in the IAV sector. No geophysical prospecting surveys exploring the complete saturated thickness of the Pliocene aquifers, other possible older underlying Neogene formations of unknown hydrogeological behavior, and the IAV basement depth and typology could be compiled. However, three surveys partially explored these features in the EAC and WAC sectors. Therefore, this basic information remains unknown, preventing an accurate assessment of the groundwater resource from which to deduce the renewable fraction of thick regional Pleistocene and Pliocene aquifers that can be exploited sustainably. Geophysical prospecting surveys with greater penetration depth could provide this basic information.

This paper demonstrates the need to systematize the use of geophysical prospecting techniques, including the most widely used technique described here to deduce shallow aquifer typologies and transient groundwater features and other specifics to explore the complete saturated thickness of Pleistocene and Pliocene aquifers forming the larger freshwater reservoirs in the MDQ. The above findings and research gaps, together with the generated database, seek to improve the design of geophysical prospecting surveys to explore groundwater resources in the MDQ and other large Andean urban areas.

**Author Contributions:** Conceptualization, F.J.A.; methodology, F.J.A.; formal analysis, L.P. and F.J.A.; data curation, L.P.; writing—original draft preparation, L.P., F.J.A. and J.S.-A.; writing—review and editing, F.J.A. and J.S.-A.; project administration, F.J.A.; funding acquisition, L.P. and F.J.A. All authors have read and agreed to the published version of the manuscript.

**Funding:** This research was partly supported by the International Atomic Energy Agency Project ECU/7/006 and the Ecuadorian Research Project PROMETEO-CEB-014-2015.

**Institutional Review Board Statement:** Not applicable.

**Informed Consent Statement:** Not applicable.

**Data Availability Statement:** The data presented in this study are available on request from the corresponding author.

**Acknowledgments:** The authors are grateful for the information on the geophysical prospecting surveys provided by the EPMAPS. We also wish to express our gratitude to two anonymous reviewers for their valuable advice and comments.

**Conflicts of Interest:** The authors declare no conflict of interest.

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
