# Peer review of "Usefulness of Compiled Geophysical Prospecting Surveys in Groundwater Research in the Metropolitan District of Quito in Northern Ecuador"

_applsci, doi:10.3390/app112311144_

Round 1

Reviewer 1 Report

Dear authors,

Please consider the following remarks.

- the title of the manuscript is too general. If the aim of the paper is to create a database of geophysical research, this should be reflected in the title.

- the structure of the article does not comply with the IMRAD format (https://www.mdpi.com/journal/applsci/instructions)

- the goal of the research, the study's objectives and scientific contributions are not clearly defined

- pages are not numbered correctly (after page 7 the numbering is reset)

- it is not clear why the results of only some surveys are presented (ERT survey 19 in Figure 2, REMI survey in Figure 3, MTS survey in Figure 4) According to which criteria were only those results selected?

- line 79 - the manuscript mentions "assessment of the fraction of groundwater that can sustainably supplement urban water demand in the MDQ". Apart from effective porosity, no other relevant hydrogeological parameter of aquifers is considered.

- line 131 - the abbreviation „DEM“ is not explained

- line 503 - which transient groundwater features of Holocene and late Pleistocene formations were explored?

Author Response

See the attached PDF file.

Reviewer 2 Report

The manuscript “Geophysical prospecting surveys for groundwater research in the Metropolitan District of Quito in northern Ecuador presents review of geophysical methods applied in groundwater research projects. The main weakness of the paper is that it only gives an overview of the performed geophysical measurements and their potential in groundwater research, which are already well known. However, the manuscript lacks a scientific contribution. The conclusions are very general and do not point to new findings. The paper is poor in novelty, discussion, and conclusions but could be improved. If the manuscript were complemented by a result produced from geophysical models such as an aquifer model or a geological model, it would gain in significance.

Title

In my opinion, the title does not fit with the paper contents. It indicates that the paper emphasise geophysical surveys and their interpretation, but it is a compilation of the results of previous geophysical surveys. I therefore suggest changing of the title.

(example: Groundwater resource evaluation in the Metropolitan District of Quito in northern Ecuador based on geophysical surveys)

Specific comments and suggestions:

Abstract:

L20: some?? Maybe would be better to use “certain”

L23: “Of the surveys,..” I suggest change to “Fifteen of the EPMAPS-promoted surveys used near-surface electrical…”

L30: change “scientific literature” to scientific papers

L34-36 “This research gap prevents the accurate assessment of the renewable groundwater fraction of the regional aquifers in the MDQ that can be exploited sustainably.” Can you suggest possible solution?

Introduction

Table 1 can be excluded. Just write in full on first use with the abbreviated form in parentheses. Some of the terms have standard abbreviations, such as electrical resistivity (R), electrical conductivity (C), shear-wave velocity (Vs) and in italic. Acronym for Refraction microtremor is more commonly written as ReMi. In the text it is better to use resistivity and conductivity instead of abbreviations.

Data Compilation

L219-223 I would’t use “reinterpretation” if it means adjusting the drawing style or standardization of description.

Results

I suggest subtitles Near-surface electrical surveys/ Near-surface seismic surveys/Electromagnetic surveys

L245: potential electrodes instead of reading electrodes

L246 current electrodes instead of additional electrodes

L248: Here, I do not see a need for references 53-55

L252: for 1D electrical resistivity distribution

L255: VES gives 1D resistivity distribution; how is it used to define the geometry of aquifer?

L259: use configuration instead of dispositions

L276: electrode array

Conclusions

L494: change literary data to published data

L494-501: Please rewrite the section

L501-502: I do not understand the sentence “Most of the surveys were performed in the IAV sector, where concentrated groundwater exploitation.

The conclusion is that electrical surveys explored aquifer geometry of Holocene and late Pleistocene formations, seismic surveys examined the geometry of Holocene formations and electromagnetic surveys Holocene to late Pliocene formations. Those conclusions are very general, and seismic methods and ERT are common methods for  subsurface mapping, and their depth of investigation is also well known. Suggestion: possibly geological models based on presented geophysical models and other available geological and geotechnical data would improve the paper. The models could supplement results and conclusions.

Language: Sentence structure needs to be improved throughout the text

Author Response

See the attached PDF file.

Round 2

Reviewer 1 Report

The authors have answered all questions and corrected the manuscript according to the recommendations of the reviewer.

Author Response

Thank you very much for your constructive review.

Reviewer 2 Report

The comments have been considered and answered. The scope of the manuscript is stated more clearly. But attention still should be paid to the following, especially to references cited. Several examples are below:

L37 suggestion: "Geophysical  prospecting  surveys with greater penetration depth are proposed..."

L84: did you mean "...concerning the aquifer saturated thickness,...

L549-550: suggestion "MTS  surveys with greater penetration depth could  provide  this  basic information." Since high-frequency magnetotelluric sounding can be associated with smaller penetration depth than low-frequency survey. Although, penetration depth or skin depth is dependent on bulk resistivity and frequency (period).

L396-398: In the sentence “One  survey  used  low-frequency  MTS  (< 10  Hz)  in  13  profiles  with  a  prospecting length in the 160–750 m range, a maximum prospecting depth of 130 m, and ER in the 280 Ω m range [39].” The reference is to PhD Thesis (Rios-Sanchez, M.  A remote sensing approach to characterize the hydrogeology of mountainous areas: Application to the Quito Aquifer System (QAS), Ecuador. PhD Thesis, Michigan Technological University, Houghton, USA, 2012.). But the thesis research uses low-frequency electromagnetic method rather than magnetotelluric method : “Vertical electrical sounding (VES), two-dimensional electrical resistivity profiling (2D-ER), and very low frequency electromagnetic method (VLF-EM) were applied to define the extent and depth of the tectonic features mapped based on the lineament analysis.” Page 67

L398-401 “Three surveys used high-frequency MTS (> 10 Hz) [18,45,52] with a total prospecting length of 33.3 km, a maximum prospecting depth of 4,000 m, and ER in the  ranges  of  10–220 Ω m  for  geological  formations  catalogued  as  aquifers  and  220–27,100 Ω m for geological formations catalogued as aquitards.” One of the surveys refers to Peñafiel, L.; Reyes, P.S.B.; Alcalá, F.J.; Ramírez, M.R.; Cabero, A. Fold-axis parallel extension along the southern ending of the  Quito (Ecuadorian Andes) fault system: Implications in river network and aquifer geometry. Geotectonics 2020, 54, 256–265. where it is stated that MT survey of low frequency is applied “Along a perpendicular  line  to  the  Saguanchi  Gorge  downstream direction,  13  arrays  (Fig.  2,  measuring  points)  were surveyed.  In  each  measurement  point,  three  low  frequencies  to  accomplish  a  penetration  depth  from  0.6 to 1.8 km were used.” Referencence 52 as well.

L409-415: similar as above

The magnetotelluric sounding is compared to VLF-EM method.

There is an extra space in Ωm (throughout the text)

Although, this is not crucial for the manuscript I believe that the text would be easier followed if abbreviations were not used so often (also, I believe there is no need for it). Regarding the resistivity abbreviation, even more common (than R) is symbol ρ and for apparent resistivity ρa, and for shear velocity Vs (or use S-wave velocity).

Author Response

The comments have been considered and answered. The scope of the manuscript is stated more clearly. But attention still should be paid to the following, especially to references cited. Several examples are below:

Thank you for your constructive review.

L37 suggestion: "Geophysical prospecting surveys with greater penetration depth are proposed..."

Done. See comment R2C1, page 1.

L84: did you mean "...concerning the aquifer saturated thickness,...

No, we refer to “general" hydrogeological knowledge, which includes many other aquifer features. Aquifer saturated thickness is one of these features, certainly an important one. See R2C2 comment, page 2.

L549-550: suggestion "MTS  surveys with greater penetration depth could  provide  this  basic information." Since high-frequency magnetotelluric sounding can be associated with smaller penetration depth than low-frequency survey. Although, penetration depth or skin depth is dependent on bulk resistivity and frequency (period).

A more generic sentence was included as “Geophysical prospecting surveys with greater penetration depth could provide this basic information”. See comment R2C3, page 17.

L396-398: In the sentence “One  survey  used  low-frequency  MTS  (< 10  Hz)  in  13  profiles  with  a  prospecting length in the 160–750 m range, a maximum prospecting depth of 130 m, and ER in the 280 Ω m range [39].” The reference is to PhD Thesis (Rios-Sanchez, M.  A remote sensing approach to characterize the hydrogeology of mountainous areas: Application to the Quito Aquifer System (QAS), Ecuador. PhD Thesis, Michigan Technological University, Houghton, USA, 2012.). But the thesis research uses low-frequency electromagnetic method rather than magnetotelluric method : “Vertical electrical sounding (VES), two-dimensional electrical resistivity profiling (2D-ER), and very low frequency electromagnetic method (VLF-EM) were applied to define the extent and depth of the tectonic features mapped based on the lineament analysis.” Page 67

Thank you for this comment. This mistake derives from an excessive simplification of the compiled database. Low-frequency MTS surveys correspond to VLF-EM surveys. The subsequent corrections are included in the text, figures 1, 4 and 5 (they were replaced), and table 2. The changes have adequately been highlighted. See comment R2C4, pages 1, 2, 5, 9, 14, 16-18.

L398-401 “Three surveys used high-frequency MTS (> 10 Hz) [18,45,52] with a total prospecting length of 33.3 km, a maximum prospecting depth of 4,000 m, and ER in the  ranges  of  10–220 Ω m  for  geological  formations  catalogued  as  aquifers  and  220–27,100 Ω m for geological formations catalogued as aquitards.” One of the surveys refers to Peñafiel, L.; Reyes, P.S.B.; Alcalá, F.J.; Ramírez, M.R.; Cabero, A. Fold-axis parallel extension along the southern ending of the  Quito (Ecuadorian Andes) fault system: Implications in river network and aquifer geometry. Geotectonics 2020, 54, 256–265. where it is stated that MT survey of low frequency is applied “Along a perpendicular  line  to  the  Saguanchi  Gorge  downstream direction,  13  arrays  (Fig.  2,  measuring  points)  were surveyed.  In  each  measurement  point,  three  low  frequencies  to  accomplish  a  penetration  depth  from  0.6 to 1.8 km were used.” Referencence 52 as well.

Thank you for this comment. As replied in the above comment, this mistake derives from an excessive simplification of the compiled database. High-frequency MTS surveys correspond to low-frequency MTS surveys. The subsequent corrections are included in the text, figures 1, 4 and 5 (they were replaced), and table 2. The changes have adequately been highlighted. See comment R2C5, pages 2, 7, 9, 14-18.

L409-415: similar as above

Corrected. See the above comments R2C4 and R2C5.

The magnetotelluric sounding is compared to VLF-EM method.

Prospecting length and depth of surveys performed in the Inter-Andean Valley sector were compared, as shown in figure 5. See comment R2C6, page 16.

There is an extra space in Ωm (throughout the text)

Really they are non-breaking spaces (Ctrl+Shift+Spacebar) used in formal edition of variables in Mathematics, Physics and Experimental Sciences scientific documents. The non-breaking spaces were premeditated for resistivity (Ω m) and shear-wave velocity (m s-1); see the multiple examples throughout the text (Word version). However, both Ωm and Ω m are indistinctly used in the scientific literature. There are many examples of both cases. The adopted notation (using non-breaking spaces) was also used in previous papers published this year in this journal.

Although, this is not crucial for the manuscript I believe that the text would be easier followed if abbreviations were not used so often (also, I believe there is no need for it). Regarding the resistivity abbreviation, even more common (than R) is symbol ρ and for apparent resistivity ρa, and for shear velocity Vs (or use S-wave velocity).

As replied in the first round of revision, acronyms ER and VS are easy to follow. They have been extensively used in the scientific literature. These acronyms were also used in previous papers published this year in this journal.